# Bioactive Phenolic Compounds from *Primula veris* L.: Influence of the Extraction Conditions and Purification

**DOI:** 10.3390/molecules26040997

**Published:** 2021-02-13

**Authors:** Maria Tarapatskyy, Aleksandra Gumienna, Patrycja Sowa, Ireneusz Kapusta, Czesław Puchalski

**Affiliations:** 1Department of Bioenergetics, Food Analysis and Microbiology, Institute of Food Technology and Nutrition, College of Natural Sciences, University of Rzeszów, 35-601 Rzeszów, Poland; gumienna.ola@wp.pl (A.G.); psowa@ur.edu.pl (P.S.); cpuchal@ur.edu.pl (C.P.); 2Department of Food Technology and Human Nutrition, Institute of Food Technology and Nutrition, College of Natural Sciences, University of Rzeszów, 35-601 Rzeszów, Poland; ikapusta@ur.edu.pl

**Keywords:** bioactive compounds, cowslip, extraction, polyphenols, antioxidant activity, health-promoting quality

## Abstract

Our experiments may help to answer the question of whether cowslip (*Primula veris* L.) is a rich source of bioactive substances that can be obtained by efficient extraction with potential use as a food additive. A hypothesis assumed that the type of solvent used for plant extraction and the individual morphological parts of *Primula veris* L. used for the preparation of herbal extracts will have key impacts on the efficiency of the extraction of bioactive compounds, and thus, the health-promoting quality of plant concentrates produced. Most analysis of such polyphenolic compound contents in extracts from *Primula veris* L. has been performed by using chromatography methods such as ultra-performance reverse-phase liquid chromatography (UPLC−PDA−MS/MS). Experiments demonstrated that the most effective extraction agent for fresh study material was water at 100 °C, whereas for dried material it was 70% ethanol. The richest sources of polyphenolic compounds were found in cowslip primrose flowers and leaves. The aqueous and ethanol extracts from *Primula veris* L. were characterized by a quantitatively rich profile of polyphenolic substances, and a high antioxidative potential. Selective extraction with the use of mild conditions and neutral solvents is the first step to obtaining preparations from cowslip primrose with a high content of bioactive substances.

## 1. Introduction

Cowslip (*Primula veris* L.) is a well-known medicinal herb. The plant has yellow flowers with a pleasant, honey smell that form a canopy at the top of an inflorescence stalk, which is 10 to 20 cm high and which grows from the leaf rosette [1]. It is used as a garden ornament, but also as a decorative component in many dishes. Flowers and leaves of this plant are used for salads, sandwiches, and desserts, among other things. In addition to its aesthetic features, it also has medicinal properties. The leaves of the cowslip contain large amounts of vitamin C and the flowers contain large amounts of flavonoids. In the past, a substitute for tea was made from its flowers [2]. It is now mainly used as infusions or liqueurs for any respiratory, cardiac, and nervous system condition due to its expectorant, sedative, constrictive, diuretic, anti-inflammatory, and antioxidative effects [3,4]. Due to its valuable chemical composition, this plant is also used in cosmetology and dermatology. The high levels of saponins and flavonoids present in cowslip make this plant stand out with the diversity of its biological activity. The scope of application of saponins is limited by their high toxicity. Nevertheless, these compounds are a great hope, among other things, in the fight against cancer [5].

At the moment, consumers are very consciously choosing less-processed products in the hope that they exhibit better health properties and precious nutritional value. In light of these behaviors, all measures should be taken to protect valuable nutrients in agricultural raw materials, particularly those undergoing processing. The analysis of the profile of polyphenolic compounds present in cowslip flowers is the answer to the question of whether these herbaceous plants can become a valuable food additive and thus support effective health-promoting disease prevention.

According to literature data, the content of polyphenolic compounds in a cowslip depends on, i.e. conditions for their extraction and variations in the different morphological parts of the plant [6,7]. An in-depth analysis of the extraction process, especially the process conducted under mild conditions, using neutral solvents and low temperatures, will allow the determination of the optimum parameters for the effective extraction of bioactive substances from *Primula veris* L. and the identification of the plant parts with their greatest concentrations, even allowing us to establish safe ranges of the use of cowslip for health care.

Currently, conventional aqueous extraction using high temperatures and a long extraction time is being overtaken by techniques supporting the efficiency of bioactive substance diffusion, such as the supercritical CO_2_-assisted extraction [8]. This method was used, inter alia, in experiments on the decaffeination of coffee and tea and the improvement of the extraction efficiency of hop oils [9]. Another way of assisting the extraction of components in an aquatic environment is microwave radiation [10,11] and the most popular method using ultrasound [12,13].

The optimization of the extraction of bioactive substances from cowslip, *Primula veris* L., taking into account the different morphological parts of the plant and the various solvents, is intended primarily to seek health-promoting food additives and to support health-promoting disease prevention and biomedicine. One of the assumptions of the experiment was to exclude solvents not allowed in food processing and to simplify the purification process to the minimum necessary for preparations to be easy to prepare. The main objective was to maximize extraction efficiency and thus the concentration of the polyphenolic compounds. When selecting extraction solvents, the recommendations of herbalist practices have been followed, according to which, for herbs, water infusions intended for consumption and ethanol liqueurs of different alcoholic strengths intended for consumption and skin applications, are most often produced.

In view of the results of the preliminary tests using fresh cowslip flowers, a test hypothesis was formed, which assumed that both the type of solvent used for plant extraction and the individual morphological parts of *Primula veris* L. used for the preparation of herbal extracts will have a key impact on the efficiency of the extraction of bioactive compounds, and thus, the health-promoting quality of the plant concentrates produced. The results of the experiments show differences in the effectiveness of the extraction of the health-promoting substances from different morphological parts of the cowslip (*Primula veris* L.), as well as the quantitative and qualitative differences of the profile of polyphenolic compounds identified in samples of the extracts. The analysis of the content levels of health-promoting substances in cowslip extracts can help to determine the level of fortification of food, and even the manner and form in which plant additives may be introduced to food, thus developing preliminary daily intake recommendations for health-promoting disease prevention.

## 2. Results

Based on the chromatographic analysis with MS detection, 18 polyphenolic compounds were identified in the examined extracts of different parts, of which the predominant group consists of quercetin and kaempferol derivatives. The names of the identified compounds are given in Table 1. Table 2 shows the concentrations of the individual polyphenolic compounds together with their total content as determined in the extracts from the cowslip flowers (*Primula veris* L.). The chromatogram of the identified phenolic compounds of *Primula veris* L. in the dried flower extracts is presented in Figure 1.

Based on the analysis of the contents of bioactive compounds in extracts made of both fresh and dried flowers of *Primula veris* L., it was found that water at 100 °C was the best extraction solvent for polyphenolic compounds contained in the fresh flowers of *Primula veris* L., while in the case of the samples extracted with ethanol, the most efficient solvent was a solution with an alcohol content of 70% and the least effective was pure 96% ethanol. It was found that, for solutions with an ethanol content of 40%, the profile of polyphenolic compounds is the least differentiated, however, the total content of the identified compounds was similar to the extracts obtained with solutions with an ethanol content of 70%. The average bioactive compound content in aqueous extracts was 628.24 mg/L, and in ethanol extracts, it was lower by 25–40%. Substances identified in the largest quantities in aqueous and ethanol extracts prepared with the fresh flowers of *Primula veris* L. were quercetin 3-*O*-rutinoside (rutin) and isorhamnetin 3-*O*-rutinoside. The rutin content was on average 205.96 mg/L in aqueous extracts and between 133.27 and 158.28 mg/L in ethanol solutions, whereas isorhamnetin 3-*O*-rutinoside in aqueous solutions was determined to be 230 mg/L and in ethanol solutions this concentration lessened by as much as 44%. When analyzing extracts prepared with dried plant obtained under laboratory conditions and commercially available dried cowslip, profiles of polyphenolic compounds were found to differ slightly in terms of quality, depending on the type of solvent. Extracts from the commercially available dried cowslip showed the widest spectrum of polyphenolic compounds compared to extracts from the dried cowslip obtained under laboratory conditions, which may be because they contain two varieties—*Primula veris* L. and *Primula elatior* (L.) Hill. In quantitative terms, the most valuable extracts have been obtained from the dried cowslip collected in 2019, which may suggest that the storage time causes degradation of the bioactive compound content. Their content in the aqueous extracts analyzed in the two harvest years was compared and varied on average by 47%. In ethanol extracts, the smallest difference of 17% between the dried herbs from the harvests in 2018 and 2019 was found in the 70% ethanol solution, and this result could suggest a percentage decrease in activity resulting from storage. The most effective extraction of bioactive substances from the dried flowers was found in the case of ethanol solutions with 70% and 40% alcohol content and hot water infusions, where the average phenolic compound content was between 1687.45 and 3591.82 mg/L. The least effective extraction was found in the case of pure ethanol with 96% alcohol content, where the content of these substances was between 1198.98 and 2843.71 mg/L. Compared to fresh flower extracts, the concentration of polyphenolic compounds in dried plant extracts increased by more than four times, which is proportional to the amount of water lost as a result of drying, and demonstrates the gentleness of this process to bioactive substances. Aqueous and ethanol extracts prepared with the dried cowslip produced under laboratory conditions—as with the fresh flower extracts—were characterized by a high content of two compounds: quercetin 3-*O*-rutinoside (rutin), with an average content of 347.93–1000.54 mg/L, and isorhamnetin 3-*O*-rutinoside in a quantity close to that of the rutin content. In turn, extracts from commercially available dried plants were characterized by having up to four times lower rutin content levels and around ten times lower isorhamnetin 3-*O*-rutinoside content levels compared to laboratory dried herb extracts, and an even lower content of this substance than in the case of extracts from the fresh flowers of *Primula veris* L. In contrast, the composition of these extracts was significantly richer in numerous quercetin derivatives in significant quantities ranging from 115 to 367 mg/L, depending on the type of solvent used for extraction, displaying up to three times higher content levels of kaempferol 3-*O*-rutinoside-7-*O*-rhamnoside compared to extracts from the dried plants obtained under laboratory conditions, and are 20 times higher content levels than in the extracts from fresh flowers.

When comparing the polyphenolic compound content of extracts from cowslip, it should be noted that aqueous solutions were prepared at a higher temperature (100 °C) than in the case of ethanol solutions (room temperature). Therefore, in the case of fresh flowers, which contained up to 87% water, the temperature of the extraction solvent was lowered more quickly than in the case of dried flowers containing 6–7.3% water, which could also have an impact on the efficiency of the bioactive substance extraction.

The second stage of the experiment consisted of the analysis of the extracts from the *Primula veris* L. flowers together with the stalk, which represented up to 30% of the sample weight. The content of the individual bioactive substances depending on the type of solvent used for extraction is given in Table 3.

The chromatographic analysis of the extracts from flowers together with stalks confirmed a significantly lower, by around 60%, polyphenolic potential, for fresh material and aqueous and ethanol extraction by around 50–60%, and for dried material by 40–45%, compared to the extracts from the flowers of *Primula veris* L. alone. Referring to extraction temperatures and the freedom of extraction from fresh material, it should be noted that the stalks not only contained fewer bioactive substances but were also rich in many structural polysaccharides stiffening this part of the plant, which may also have affected the extraction efficiency [14]. The profile of polyphenolic compounds in the extracts from inflorescences and stalks was similar to that of the extracts containing only flowers of *Primula veris* L. The predominant compounds in these extracts were quercetin 3-*O*-rutinoside (rutin) and isorhamnetin 3-*O*-rutinoside, and their content was about 30–60% lower than in the extracts which were only from the cowslip flowers. The highest polyphenolic compound content of 1999.26 mg/L was determined in aqueous extracts from dried test material, and in aqueous extracts from fresh material where the content of these substances was 259.77 mg/L. The least effective solvent used for extraction was pure ethanol with an alcohol content of 96% and the total polyphenolic compound content in the samples with this solution was approximately 30% lower than in the aqueous extracts. The average concentration of the bioactive substances after drying was similar as in the case of the flowers of *Primula veris* L. alone. In order to confirm the efficiency of extraction of the selected solvent group and to determine the levels of polyphenolic compounds from *Primula veris* L. in the respective morphological parts of cowslip, extracts from the other above-ground parts, that is stalks (Table 4) and leaves (Table 5), were taken.

In the case of extracts from fresh stalks, the total polyphenol content ranged from 39.23 to 97.07 mg/L, and the dried material extracts showed polyphenolic activity at the levels of 422.31 to 886.31 mg/L. Extracts with 40% and 70% ethanol were characterized by the most effective extraction, and the least effective were extracts with 96% pure ethanol. Compared to the results in Table 3, it was found that the stalks do not constitute a good source of polyphenolic compounds in cowslip in both quantitative and qualitative terms. In aqueous samples from the *Primula veris* L. stalks alone, a polyphenolic compound content of 25–27% was obtained compared with the extracts from stalks and inflorescences, while the fresh and dried ethanol extracts only obtained 22–48% of this level. In turn, for *Primula veris* L. leaf extracts, a much higher polyphenolic potential was observed than in stalks, but their qualitative profile was poorer compared to the samples of extracts made of flowers alone. The range of concentrations of polyphenolic compounds for fresh material was from 286.36 mg/L in samples extracted with pure ethanol 96% to 319.14 mg/L in samples extracted with 40% ethanol, which proved to be the best solvent for the extraction of bioactive substances from the leaves of *Primula veris* L. The efficiency of the aqueous extraction was similar to that achieved for extracts made with 40% ethanol. The substance extracted from leaves in the largest quantity was kaempferol 3-*O*-rutinoside-7-*O*-rhamnoside, with a content of up to 425.56 mg/L in the samples of dried leaves and 70% ethanol. This compound can be considered as indicative for leaves, since no such high concentrations of this substance have been detected in any of the previously analyzed extracts from the flowers and stalks of *Primula veris* L. The second key compound in the dried leaf extracts in terms of quantity was quercetin 3-*O*-rutinoside (rutin), the content of which was between 255.98 and 319.64 mg/L and was approximately 40% lower than in the extracts from inflorescences with stalks. As in previous observations, pure ethanol 96% proved to be the weakest extraction solvent and the lowest concentrations of polyphenolic compounds were obtained in the samples containing it. The drying process in laboratory conditions for leaves has resulted in a fourfold increase in bioactive compounds. At this stage of the analysis, it can be concluded that the dried *Primula veris* L. materials obtained under gentle conditions constitute a very valuable, concentrated source of polyphenolic compounds without loss of their activity and that this method can be an effective way of preserving these plants.

In the last step of the study, the extracts from the roots of cowslip, fresh and dried under laboratory conditions, were prepared, and the results of the analyses are shown in Table 6. It appears from scientific literature [15] and pharmacopoeia papers [16,17] that these underground parts of cowslip are used in pharmaceutical preparations with expectorant effect due to their high levels of saponins, mainly Primula saponin II, Primula saponin I, and Priverosaponin B 22-acetate; therefore, it was expected that polyphenolic activity in the roots will be negligible compared to the other morphological parts of *Primula veris* L. Table 6 shows the average results of the polyphenolic compound content in aqueous and ethanol extracts from the fresh and dried roots of *Primula veris* L.

The analysis of the profile of polyphenolic compounds in root extracts confirmed the low levels of polyphenolic compounds among all the morphological parts of cowslip examined. In aqueous and ethanol solutions with the lowest alcoholic strength, the predominant compounds were kaempferol 3-*O*-rutinoside-7-*O*-rhamnoside (0.52–29.16 mg/L) and quercetin 3-*O*-rutinoside (rutin) (1.38–17.32 mg/L). In the case of aqueous extractions, drying resulted in a double increase in the concentrations of polyphenolic components in the extracts and, in the case of ethanolic solutions, more than a fourfold increase in the concentration of these compounds. The highest polyphenolic compound content determined in the dried roots extracted with 70% ethanol was 122.46 mg/L and was over seven times lower than the content of these substances in the dried cowslip stalks, for which this solvent was also the most effective. The least effective solvent was pure ethanol with an alcohol content of 96%, and a total polyphenolic compound content of 9.86 mg/L was found in samples containing it.

Table 7 shows the results of the statistical analysis of the influence of the plant part of fresh *Primula veris* L. and extraction solution on the total polyphenolic compounds and antioxidant properties in the extracts. In turn, Table 8 shows the results of the statistical analysis of the influence of the different morphological parts of cowslip and the type of solvent used to extract the polyphenolic substances from the dried test material.

The statistical analysis showed very significant differences between the total polyphenolic compound contents in the individual parts of the fresh plant of *Primula veris* L. irrespective of the type of solvent used for extraction. The most polyphenolic compounds were identified in the cowslip flowers and leaves, and the least in the stalks and roots. The average polyphenolic compound content in roots, irrespective of the type of solvent used for extraction, was more than 22 times lower than in the flowers and 14 times lower than in the leaves of cowslip. The analysis of the antioxidant potential by the FRAP and ABTS methods confirmed the highest activity for the flowers and leaves of cowslip, while the results of the determinations using the DPPH method showed that extracts from fresh leaves of *Primula veris* L. were characterized by the highest antioxidant potential. Significant differences between average results were also confirmed within the individual solvents used for the extraction of polyphenolic compounds. The highest efficiency was obtained for aqueous extracts (total HPLC) and for ethanol solutions of 40% (DPPH, FRAP). The lowest efficiency was observed for pure ethanol with an alcohol content of 96% confirmed by both the lowest total number of compounds identified using the HPLC technique and the lowest antioxidant potential determined by other methods.

In the case of extracts from dried parts of *Primula veris* L., the statistical analysis also confirmed the very significant differences between the total polyphenolic content and the antioxidant potential between the different morphological parts of cowslip used for extraction. The highest content of these substances was found for the flowers and inflorescences with stalks, and the lowest for dried roots of *Primula veris* L. In addition, more than a fivefold increase in the concentrations of these substances in the extracts from dried flowers has been confirmed compared to fresh flowers. The statistical analysis of the extraction efficiency indicated that ethanol solutions with an alcohol content of 70% were the most effective solvent for extracting polyphenolic compounds from the dried morphological parts of *Primula veris* L. In the case of this substance, the highest concentration of polyphenolic compounds was also obtained at a level that was almost seven times higher compared to the samples from fresh plants. The least effective solution for extraction was pure ethanol with a strength of 96%, as in the case of the fresh material results.

The antioxidant capacity of the tested extracts was highly correlated with the levels of the phenolic content obtained both by TPC and HPLC, regardless of the extraction method used (r = 0.961 for TPC and DPPH, r = 0.974 for TPC and FRAP, r = 0.937 for TPC and ABTS, r = 0.929 for HPLC analysis and DPPH, r = 0.955 for HPLC analysis and FRAP, r = 0.941 for HPLC analysis and ABTS). This confirms previous observations that phenolic compounds are largely responsible for the antioxidant activity [18]. High correlation between TPC and the total phenolic content obtained by the HPLC method was observed (r = 0.94), as well as between antioxidant activity measurement by different tests (r = 0.961 for DPPH vs. FRAP, r = 0.977 for DPPH vs. ABTS and r = 0.947 for ABTS vs. FRAP). This confirms the correctness of the methods performed.

The results of the tests carried out as part of this work were also analyzed using the hierarchical clustering analysis and heatmap visualization to determine the relationship between the examined cowslip extracts based on the tested parameters, i.e., the phenolic compound content and the antioxidant activity (Figure 2) for the different morphological parts of the plant as well as the different extraction solvents. The analysis was performed using the Euclidean distance as the measure of distance and Ward’s method as the method of merging objects. The analyzed variables had a different unit, so the standardization of values was made. Finally, a color scheme (heatmap) was applied for the visualization and the data matrix is displayed. Based on the color scale in the heatmap the values of the individual parameters can be compared (where darkest red means the highest value of a given compound content or antioxidant activity and the darkest green means the lowest value). The samples with the most similar values of the designated parameters are located closest to each other. The samples examined were divided into two main clusters. The highest antioxidant activity and the overall phenolic compound content were characteristic of the extracts of dried flowers, in particular Dried3 (dried flowers collected in 2019) flowers (located in one cluster, irrespective of the extraction solvent used), as well as dried leaves and flowers with stalks. In general, the part of the plant from which the extract was obtained was more important in the classification of the samples than the type of solvent used. The exception was extraction using 96% ethanol, where significantly lower values of the tested parameters were obtained and the extracts were located in separate, often remote clusters, such as was the case in the dried leaf extract. Differences in the content of the various phenolic compounds have been demonstrated. For example, for dried leaf extracts, the high content of kaempferol 3-*O*-rutinoside-7-*O*-rhamnoside is characteristic, while the flowers of Dried1 (commercial sample) were characterized by a high content of quercetin derivatives (in particular, quercetin 3, 7, 4′-*O*-triglucoside, quercetin 3-*O*-rutinoside-7-*O*-rhamnoside, quercetin 3-*O*-diglucoside-7-*O*-glucuronide, and quercetin 3-*O*-glucoside). Extracts with low values of the parameters tested—the extracts from fresh parts of plants and dried stalks and leaves—were located in the second main cluster. The extracts from the fresh roots showed particularly low values of the tested parameters. The multidimensional statistics allows for a fast indication of which of the extracts examined contain the highest levels of bioactive compounds, and whether they exhibit the highest antioxidant activity.

## 3. Discussion

More than 4000 chemically unique, low-molecular-mass compounds classified as flavonoids have been identified in the plant material. Flavonoids may be present in plants in a free state, as aglycones, or more frequently, in the form of sugar bonds, mainly β-glycosides (except for catechins). Glycosylation of natural compounds is generally considered to be a process aimed at increasing their solubility, and thus, facilitating intracellular and intercellular transport. On the other hand, glycosylation can be interpreted as a process of deactivating biologically active aglycone and protecting cell organelles from damage. An example may be quercetin, which is an aromatic hydrophobic compound, and its water solubility increases with the addition of further sugar residues. This activity is inversely proportional to the antioxidant properties that decrease in the presence of sugar substituents [19].

Examples also include lucerne saponins, the simplest compounds of which, i.e., aglycone, the mediagenic acid and its glucoside, are the most bioactive. Further glycosylation leads to a significant reduction in anti-fungal, hemolytic, phytotoxic, and antibacterial activities [20]. When risks from pathogens appear, the enzyme system of the plant can quickly conduct hydrolysis of more complex compounds with limited activity into more active monoglucosides or even to aglycone. The transformations of flavonoid compounds affect not only the biochemistry and physiology of the plants, acting as antioxidants and enzyme inhibitors, but also as substances with beneficial properties for humans, affecting certain aspects of metabolism; therefore, their presence in our daily diet is extremely important.

The daily intake of these phytochemicals at a level of 1–2 g may provide a pharmacologically significant concentration in tissues and body fluids [21,22]. The increase in consumption of these bioactive, health-promoting substances can be achieved, inter alia, by enriching food in plant extracts with their confirmed high concentration.

The World Health Organization (WHO) states that one of the main sources of biologically active substances used in the treatment and strengthening of the immune system in various diseases is herbal products [23]. Unfortunately, in many countries, herbal medicines and preparations are not regulated as widely as conventional drug treatments, which gives a lot of freedom to both their manufacturers and consumers [24]. However, it should be noted that the use of herbal preparations in an inappropriate/uncontrolled manner may have many effects that are adverse and even dangerous to health, as suggested by numerous reports on the occurrence of allergies initiated, e.g., by flavonoids or a reduction in the absorption of iron, vitamin C, folic acid, and even antithyroid, estrogen, or abortifacient effects; it is therefore important to accurately identify both their effects and activity levels [25,26,27]. The quality, production, and processing requirements for plant material with high biological activity, as a pharmaceutical raw material, are clearly defined in the elaborations of Pharmacopoeia [16,17]. In turn, the use of herbs and extracts from plants with high bioactive potential in food technology besides the possible level of toxicity is not limited, given that, for example, very often the expected bioactivity of food is lost as a result of technological processes, heat treatment, or storage. However, it is advisable to introduce additional regulations in this area, such as the development of effective methods of efficient and selective extraction to exclude the presence of substances that are undesirable, anti-nutritional, or even toxic, and to enhance the health-promoting characteristics of herbal preparations intended for use in food. The optimization of the process of extraction of bioactive substances from cowslip as a raw material with a rich and underestimated polyphenol potential, and the analysis of the profile of health-promoting compounds in extracts in terms of their use for food fortification, is a step in the area of the above-mentioned recommendations.

Cowslip (*Primula veris* L., syn. *P. officinalis* Hill) and oxlip (*Primula elatior* (L.) Hill) are small, long-lasting perennials from the family Primulaceae, growing wild in Europe and Asia [28]. The use of cowslip (*Primula veris* L.) in both folk medicine and as a food additive is known in the countries of southeast Europe [29]. In turn, in central Europe, it is only used in a few pharmaceutical forms and food supplements as part of herbal blends with biological and pharmacological activity confirmed in the scientific and medical literature [30]. In the current edition of the European Pharmacopoeia, these plants are listed as the source of *Primula* roots, from which bioactive substances are obtained for drug use with expectorant, anti-inflammatory, diuretic, antimicrobial, antifungal, and sedative effects [17].

Due to the great species and genetic diversity and the importance of climatic, soil, and geographical conditions that determine the content of bioactive substances in *Primula veris* L., it is extremely difficult to clearly define the profile of the health-promoting substances in cowslip [31]. Many authors of previous studies have focused mainly on the roots of the plant *Primula veris* L., having regard for the potential concentration of bioactive substances in these morphological parts [32]. Other researchers, based on newer analytical techniques, question the previously defined profiles of bioactive compounds from *Primula veris* L. [33], and even demonstrate that the content of these substances in the various aerial parts of this plant may be affected by UV-B radiation [34].

According to our previous studies, the flowers of *Primula veris* L. are a very valuable research material rich in bioactive substances and their extraction does not require complex preparation and extraction techniques. However, the choice of extraction technique and solvent type is, in addition to the quality of the plant material, the key element of the selective separation of bioactive substances with the possibility to limit or exclude undesirable or harmful compounds. By analyzing the effectiveness and popularity of the techniques used to extract phenolic compounds from plant (herbal) material in the available literature, it was concluded that the highest efficiency was achieved, among other things, through the application of microwave radiation, especially at 110 °C, and ethanol with a strength of 90% for powdered plant material [35], and also by the Soxhlet method, which, unfortunately, despite the high extraction efficiency for flavonoids, was characterized by a long extraction time of up to 6 h. Weihua et al. [36] obtained the best results of the extraction of phenolic compounds in 30 min using ultrasound-assisted extraction with 70% methanol, due to which they determined a broad spectrum of bioactive compounds in the herbal material, and the efficiency of this process was much better than in the case of the Soxhlet method. In turn, Scalia et al. [37] and Sandvoss et al. [38] described in their papers effective extraction methods with simultaneous purification of the samples with a suitable filler mixed with a solvent, and the efficiency of these methods was not only higher than that of the Soxhlet method but also that of the technique with the use of ultrasonication; this method, however, is more often used to extract saponins than flavonoids and was more complicated. For the latter, the use of ultrasound to support the extraction is a simple, fast, and sufficiently efficient process with the possibility to apply any solvent with the highest extraction efficiency. The authors of the study on optimizing the ultrasound-assisted extraction of polyphenols from fresh wheatgrass had a similar opinion [39]. They examined the effects of different extraction techniques and also the effects of the solvents on the yield of extractive substances and antioxidant activity. Their research results confirmed that the ultrasound-assisted extraction technique and ethanol as the solvent gave the highest yield of extractive substances. Savic Gajic et al. [40] proved that the ultrasound-assisted extraction gave higher total phenolic content and better antioxidant activity with shorter extraction time and reduced solvent consumption, moreover, the use of lower temperatures prevents the thermal degradation of bioactive compounds in the extract. An optimal method to combine the extraction and purification techniques is the use of ultrasound and solid-phase extraction (SPE), which provide the best extraction efficiencies and allow an easy, quick, and selective method to clean the test sample and even concentrate it if required. Tarapatskyy et al. [18] used a solid-phase extraction technique in their research. They thus purified wine samples enriched with the fresh flowers of *Primula veris* L. and macerates prepared with 40%, 70%, and 96% ethanol with the addition of cowslip. They also claimed that in the case of extraction of polyphenolic substances from the fresh flowers of *Primula veris* L. the strength of ethanol reduces the total content of these substances in the extracts and their profile, which is consistent with the studies obtained in this work. They demonstrated that, compared to wines enriched with cowslip at the same level as in macerates prepared with 40% ethanol, the total polyphenolic compound content was on average 6–14 times smaller.

When it comes to the effectiveness and popularity of the solvents used by other authors in their studies on polyphenolic compound extraction, the most popular solvents were methanol with a 70% strength and ethanol, ethyl acetate and n-hexane, dichloromethane [7,41,42].

In turn, Müller et al. [43], in their studies covering methanolic extracts derived from the dried roots and flowers of two *Primula* species, showed the presence of five bioactive compounds, including three saponins and two phenolic glycosides, of which the predominant component was primeverine, found in the aerial parts of the test plant, while in the root extracts saponins were predominant, mainly priverosaponin B-22-acetate, which, according to the authors of these studies, confirms previous reports on the profile of saponins from the various morphological parts of *Primula veris* L. Similar studies were performed by Bączek et al. [4] by comparing the raw materials of wild *Primula veris* L. and *Primula elatior* (L.) Hill in terms of the profile of phenolic compounds and their concentrations using the HPLC-DAD method. The results of their analyses confirmed that the flowers of both species are rich in flavonoids, but *Primula veris* L. was characterized by a significantly higher content of isorhamnetin-3-*O*-glucoside, astragalin, and (+)—catechin, whereas *Primula elatior* proved to be a richer source of rutoside and isorhamnetin-3-*O*-rutinoside. The authors of the studies also pointed out that both species were characterized by a high rutinoside content in the range of 630.83 to 1025.96 mg/100 g dry weight, which is known to exhibit numerous anti-inflammatory, anti-oxidative, and anti-bacterial properties. In turn, our own studies showed twice the rutinoside content in the extracts analyzed compared to Bączek et al. [4], who additionally confirmed the presence of phenolic glycosides (primeverine and primulaverine) only in the plant’s roots and not in aerial parts as other authors of research had determined [43], and their content was approximately ten times higher in *Primula veris* L. compared to the underground parts of *Primula elatior*. In the summary of the observations carried out, the same authors concluded that both *Primula* species were different in content and composition of phenolic compounds and that the substances most differentiating both species could be useful chemical markers for the identification and evaluation of these species. Fico et al. [7] reached the same conclusions, indicating in their studies the differences in the flavonoid profiles for the different morphological parts of three alpine *Primula* species as the so-called morphological and phytochemical markers that differentiate the species during the studies. The studies of Lupitu et al. [6], who compared the contents of polyphenols and antioxidant activity in ethanolic extracts from the flowers, leaves, and roots of *Primula veris* L. obtained by 7-day maceration, showed that the highest bioactive substance content was in the flowers, on average between 133 and 219 mg GAE/L, followed by the roots and leaves at a similar level from 131 to 168 mg GAE/L. The substances dominant in the ethanolic extracts studied by these authors, determined with ultra-high performance liquid chromatography, were gallic acid, quercetin, and kaempferol, the most of which were in flowers and leaves, and the least in roots. According to Wichtl [28], the total content of flavonoids in the flowers of cowslip is about 3% and the substances present in the flowers in the largest amounts are: rutoside, kaempferol-3-rutinoside, and isorhamnetin-3-glucoside. The compounds identified so far in the extracts of *Primula veris* L. using the techniques of LC-MS and HPLC were: quercetin, quercetin-3-*O*-rutinoside, quercetin-3-*O*-gentiobioside, quercetin-trihexoside, kaempferol, kaempferol-3-odiglucoside-7-*O*-glucoside, kaempferol-3-rutinoside, kaempferol-3-*O*-galactoside-rhamnoside-7-*O*-rhamnoside, luteolin, isorhamnetin, isorhamnetin-3-*O*-glucoside, isorhamnetin-3-*O*-rutinoside, limocitine-3-*O*-glucoside, limocitine-3-orutinoside, apigenin, catechin, epicatechin, and epigallocatechin, as well as some methoxylated flavones [44,45,46]. However, different methods were used and the results are expressed in different units, hence direct comparison of the results is impossible.

According to Teng et al. [47] isorhamnetin aglycon reveals cytotoxic activity toward human hepatocellular carcinoma cells. In our study, the presence of this substance was also confirmed in each morphological part of the *Primula veris* L. tested, and its highest content reached 1127.71 mg/L in the ethanol extracts from dried flowers.

In the studies of Latypova et al. [3], particular attention was paid to the identification of the raw material composition of the *Primula veris* L. The solid herbal extract, the quantitative composition of which was not given, was the object of the analysis. As part of the preliminary preparations, the authors of the studies performed a selective extraction of the bioactive compounds from *Primula veris* L. using 40% hydrated ethanol, and then carried out a multi-step process of purification of these extracts in the deposit, together with the standardization of their polyphenolic composition. Their therapeutic effect on the myocardial contractile function in animals with experimental chronic heart failure (CHF) was then examined. The authors of these studies showed that the solid herbal extract obtained from *Primula veris* L. contained flavonoid aglycons, flavonoid glycosides, and polymethoxylated flavonoids, and that the extract had positive effects on the suppression of the disease induced in the laboratory animals tested. It was also confirmed that the tested herbal agent at a dose of 30 mg/kg exerts a cardioprotective effect, which is evidenced by a smaller number of animal deaths, a lower level of CHF plasma markers, and a higher increase in myocardial contraction and relaxation rates as compared to the control group.

Given the above, it can be concluded that the contents of the biologically active substances in *Primula veris* L. have not been clearly defined and still give hope for new uses of the health-promoting substances contained in this plant. Due to the diversity of the polyphenolic compounds contained in the aqueous and ethanol extracts analyzed, which were prepared using the respective morphological parts of cowslip, the focus should first be on the analysis of the areas of highest polyphenolic activity (flowers and leaves), and the levels of undesirable substances (saponins) or harmful substances, including possible toxic elements, which may be found in the extracts should be determined. Further research activities should aim at a clear determination of the ranges of food fortification with extracts or concentrates of cowslip that could strengthen disease prevention.

## 4. Materials and Methods

### 4.1. Chemicals and Reagents

Analytical grade reagents (analytical standard) intended for liquid chromatography were used for the determination: Acetonitrile CHROMASOLV^®^ gradient grade, ≥99.9% (Honeywell, Seelze, Germany), and methanol (Mallinckrodt Baker B.V., Deventer, The Netherlands). Analytical standards for chromatography, 2,2-azino-bis(3-ethylbenzothiazoline-6-sulfonic acid (ABTS), 6-hydroxy-2,5,7,8-tetramethylchroman-2-carboxylic acid (Trolox), 2,4,6-tri(2-pyridyl)-s-triazine (TPTZ), 2,2-diphenyl-1-picrylhydrazyl (DPPH), iron (III) chloride-6-hydrate (FeCl_3_), potassium persulfate (K_2_S_2_O_8_), and Folin–Ciocalteu reagent were purchased from Sigma Aldrich (Steinheim, Germany). Sodium carbonate (Na_2_CO_3_), ethanol, hydrochloric acid, and formic acid came from Poch (Gliwice, Poland). Deionized water from the deionizer, type HLP 5P was used (Hydrolab, Poznan, Poland).

### 4.2. Plant Material

The test material was the individual morphological parts of the cowslip plant (*Primula veris* L.) harvested at the beginning of April in the year 2019, in the areas of organic herb crops in the Podkarpacie region (49°40′37.1″ N 21°27′49.6″ E). The test material was washed, dried, and divided into: flowers, flowers (inflorescences) with stalks, stalks, leaves, and roots. A part was separated from each group of material divided by morphological characteristics for drying, while fresh material was frozen at −20 °C for 24 h to facilitate the grinding process in the mill and to loosen the tissue before extraction. Material intended for drying was spread out in a thin layer on separate sheets of filter paper to facilitate water absorption. Drying was carried out in laboratory conditions at room temperature in a shaded, dry, ventilated room for 6 days. The water content in the dried material was then determined as an indicator of the completion of the drying process. Bearing in mind our previous study including the extraction from dried tea [48] and the high heterogeneity of the tested dried plant material, a range of 6 to 8% of water content was assumed to be sufficient to stabilize the test material and complete the drying process. Moreover, the research material was the flowers from *Primula veris* L. collected in 2018, dried under the same conditions as the plant samples from 2019. Commercially dried herbs intended for the preparation of infusions (*n* = 3) bought in shops with organic food, containing a mixture of crushed flowers of the following two varieties of cowslip, were also examined: *Primula veris* L. and *Primula elatior* (L.) Hill. Commercially dried herbs and fresh samples were comparative material with dried herbs obtained under laboratory conditions and, therefore, their water content was also determined directly before the extraction. Furthermore, commercially dried herbs and the flowers from *Primula veris* L. collected in 2018 are used for descriptive statistical comparisons for flowers in Table 2.

### 4.3. Extraction Conditions and Purification

Fresh batches of the test material in frozen form as well as dried portions of *Primula veris* L. were (before extraction) ground in an IKA type A 11 Basic Analytical Mill (Königswinter, Germany), and then 1 g of the ground material was weighed and transferred to extraction tubes and covered with a suitable solvent in a quantity of 20 mL, sealed, and placed in the Sonic 22 ultrasonic bath from Polsonic (Warsaw, Poland) with the thermostat function for 30 min at 40 °C. The following solvents were used for the extraction: deionized water at 100 °C and ethanol at a 40%, 70%, and 96% (*v*/*v*) alcohol content. After the completion of the ultrasound-assisted extraction, the samples were transferred to the Biosan ES-20/60 rotary shaker (Józefów k/Otwocka, Poland) and mixed under similar conditions as before, for 30 min at 40 °C at 180 revolutions per minute. After the extraction using the shaker, the samples were filtered under reduced pressure on filter papers placed on a Buchner filter, ensuring that the extraction material was thoroughly dried from the solvent. The filtrate was then centrifuged in the laboratory centrifuge (Eppendorf 5702, Hamburg, Germany) for 10 min, RCF = 2600 g. After centrifuging, the supernatant was poured into separate clean tubes. Immediately before further analysis, the extracts were filtered through PTFE Merck Millipore thimble filters (Burlington, MA, USA) with a pore diameter of 0.45 µm and diluted as needed.

### 4.4. Determination of Polyphenolic Compounds

The analysis was performed according to the method described by Kapusta et al. [49]. Polyphenolic compounds were analyzed using UPLC−PDA−MS/MS Waters ACQUITY system (Waters, Milford, MA, USA), consisting of a binary pump manager, sample manager, column manager, PDA detector, and tandem quadrupole mass spectrometer (TQD) with electrospray ionization (ESI). The separation was carried out using a BEH C18 column (100 mm × 2.1 mm i.d., 1.7 μm, Waters) kept at 50 °C. A mobile phase consisting of 0.1% formic acid in acetonitrile (B) and 0.1% formic acid in water (A) was used for the separation. The gradient program was set as follows: 0 min 5% B, from 0 to 8 min linear to 100% B, and from 8 to 9.5 min for washing and back to initial conditions. The injection volume of the samples was 5 μL (partial loop with needle overfill) and the flow rate was 0.35 mL/min. The following parameters were used for TQD: capillary voltage 3.5 kV; con voltage 30 V in positive and negative mode; the source was kept at 250 °C and desolvation temperature was 350 °C; con gas flow 100 L/h; and desolvation gas flow 800 L/h. Argon was used as a collision gas at a flow rate of 0.3 mL/min. The polyphenolic detection and identification were based on specific PDA spectra, mass-to-charge ratio, and fragment ions obtained after collision-induced dissociation (CID). The quantitative analysis was based on specific MS transitions in multiple reaction monitoring (MRM) mode. Quantification was achieved by the injection of solutions of known concentrations ranging from 0.05 to 5 mg/mL (R^2^ ≤ 0.9998) of phenolic compounds as standards. All determinations were performed in triplicate and expressed as mg/L. Waters MassLynx software v.4.1 (Waters, Milford, MA, USA) was used for data acquisition and processing.

### 4.5. Analysis of Antioxidant Activity and Total Phenolic Compounds

Antioxidant activity was measured by three different methods: FRAP [18], DPPH [50], and ABTS assay [51]. In addition, the content of total phenolic compounds (TPC) was investigated by the Folin–Cocialteu method as described by Stratil et al. [52]. The values of antioxidant activity determined by FRAP, DPPH, and ABTS methods are expressed as mmol of Trolox equivalents per 1 L of tested extracts (mmol TE/L). The results of the total polyphenol content are expressed as 1 mg of gallic acid equivalents per 1 L (mg GAE/L) of tested extracts. All measurements were performed using Spectrophotometer UV-VIS Metash UV 5100 (Shanghai Metash Instruments, Shanghai, China) with MetaSpec Pro software (Shanghai Metash Instruments, Shanghai, China).

### 4.6. Analysis of Water Content

The water content of the samples of cowslip was determined using a moisture analyzer with an infrared emitter Ohaus MB12 (Parsippany, NJ, USA) in accordance with the standard [53].

### 4.7. Statistical Analysis

All of the analyses were made in three independent replications for each sample. The results are presented as the arithmetic mean ± standard deviation (SD). The acquired findings were subjected to statistical analyses with the use of Statistica 13.1 software (StatSoft, Inc., Tulsa, OK, USA). The significant differences between the mean values were obtained by one-way analysis of variance (ANOVA) followed by Duncan’s multiple ranges (*p* < 0.01; *p* < 0.05). Commercially dried herbs and the flowers from *Primula veris* L. collected in 2018 were used for descriptive statistical comparisons for flowers in Table 2. The flowers from *Primula veris* L. collected in 2018 were used also for statistical comparisons as mean value analysis of variance (ANOVA).

Hierarchical clustering analysis and heatmap visualization was applied to explore the similarity between the *Primula veris* L. extracts based on phenolic compound content and antioxidant activity. Clustering was performed using the Ward distance matrix that was formed based on the Euclidean distance. The correlation between the content of the compounds analyzed was determined using a Pearson’s correlation test.

## 5. Conclusions

Considering pro-health uses of cowslip primrose extracts, selective extraction with the use of mild conditions and neutral solvents is the first step to obtaining preparations with a high content of bioactive substances.

Compared to numerous results of tests using methanol to extract polyphenolic substances from *Primula veris* L., the obtained aqueous and ethanol extracts were characterized by a quantitatively similar profile of polyphenolic substances, and a high antioxidative potential. The dried material used in the study provided a considerably higher extract bioactivity compared to the fresh material, therefore, drying may be considered an effective method of preserving the *Primula veris* L. plant and its high bioactivity. Experiments demonstrated that the most effective extraction agent for fresh study material was water at 100 °C, whereas for dried material it was 70% ethanol. The lowest extraction effectiveness regarding both fresh and dried cowslip primrose was obtained with pure 96% ethanol. The dominant substances in the polyphenol profile identified in the extracts from various morphological parts of *Primula veris* L. included quercetin 3-*O*-rutinoside (rutin) and isorhamnetin 3-*O*-rutinoside; moreover, 16 other polyphenolic compounds at different concentrations were found in individual morphological parts of *Primula veris* L. The richest sources of polyphenolic compounds were found in cowslip primrose flowers, as well as in flowers with stalks and leaves, in which the polyphenolic compound content was approximately half of that detected in flowers. The study does not exhaust the methods to optimize the extraction of bioactive substances from cowslip primrose, as previous studies demonstrated that slightly different conditions and the application of maceration in low-alcohol extraction agents for a few days can result in a different spectrum of polyphenol compounds. Therefore, a potential extension of the range of extracted substances, as well as the determination of the levels of any undesirable substances, i.e., saponin, is required. At the next stage, researchers should focus on obtaining extracts at maximum concentrations, with high bioactivity and stable polyphenol contents.

## Figures and Tables

**Figure 1 molecules-26-00997-f001:**
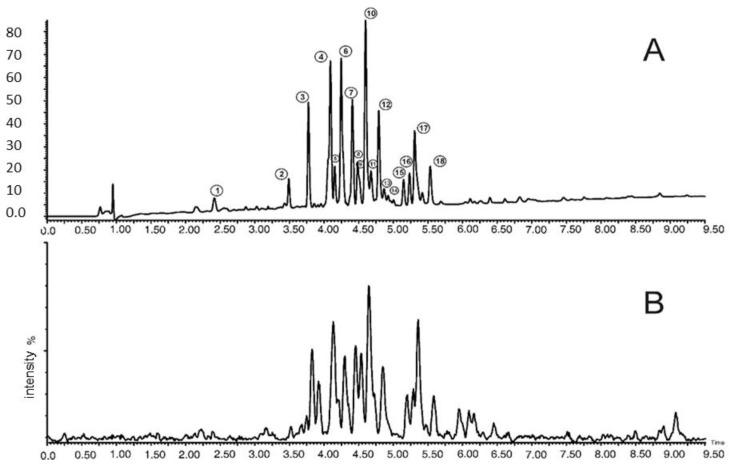
UPLC−PDA−MS/MS chromatogram of phenolic compounds of *Primula veris* L. dried flower extracts. **A**: PDA chromatogram at 254 nm; **B**: MS base peak chromatogram.

**Figure 2 molecules-26-00997-f002:**
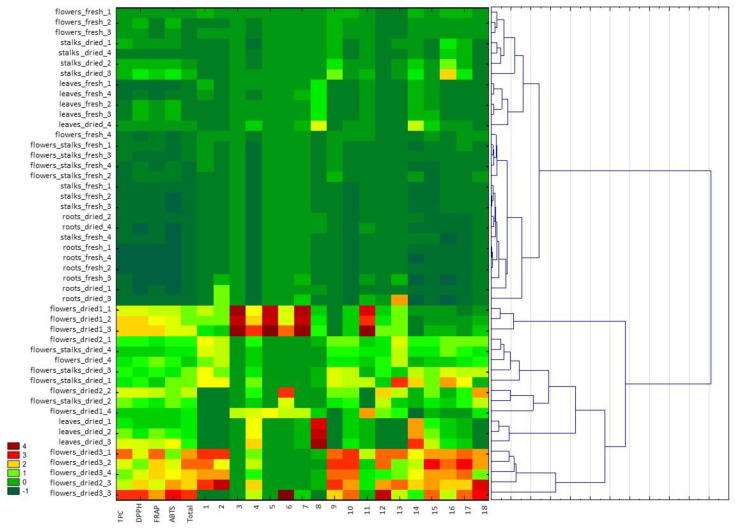
Hierarchical clustering analysis and heatmap visualization of *Primula veris* L. extracts based on the antioxidant activity determined by FRAP, DPPH, ABTS, the total content of phenolic compounds (TPC), the sum of the contents of phenolic compounds determined by UPLC−PDA−MS/MS (total), and the content of the individual phenolic compounds determined by UPLC−PDA−MS/MS (marked as **1**–**18** in accordance with Table 1). The darkest red color on the heat map represents the highest content of a particular phenol compound or the highest antioxidant activity, whereas the darkest green represents the low value of these parameters. Cluster analysis was performed using standardized data.

**Table 1 molecules-26-00997-t001:** Individual phenolic compounds identified by ultra-performance reverse-phase liquid chromatography (UPLC−PDA−MS/MS) in *Primula veris* (L.) extracts.

Compound		Rt	λ_max_	[M − H] m/z	
		min	nm	MS	MS/MS
**1**	5-*O*-(E)-caffeoyl-galactaric acid	2.46	293	371	209
**2**	Dicaffeoyl-protocatechuic acid diglucoside	3.42	260, 350	801	447, 323, 144
**3**	Quercetin 3, 7, 4′-*O*-triglucoside	3.77	255, 359	787	301
**4**	Quercetin-3-*O*-diglucoside	4.1	255, 354	625	301
**5**	Quercetin 3-*O*-rutinoside-7-*O*-rhamnoside	4.17	255, 357	755	609, 301
**6**	Quercetin 3-*O*-rutinoside-7-*O*-glucoside	4.25	255, 354	771	609, 301
**7**	Quercetin 3-*O*-diglucoside-7-*O*-glucuronide	4.4	253, 357	801	625, 301
**8**	Kaempferol 3-*O*-rutinoside-7-*O*-rhamnoside	4.47	264, 347	739	593, 285
**9**	Quercetin 3-*O*-glucoside-7-*O*-rhamnoside	4.52	255, 354	609	463, 301
**10**	Quercetin 3-*O*-rutinoside (Rutin)	4.6	255, 353	609	301
**11**	Quercetin 3-*O*-glucoside	4.73	255, 352	463	301
**12**	Quercetin 4′-*O*-glucoside	4.79	255, 352	463	301
**13**	6,3′-dimetoxyquercetin 7-*O*-diglucoside	4.87	255, 353	669	507, 345
**14**	Kaempferol 3-*O*-glucoside-7-*O*-rhamnoside	4.93	264, 347	593	447, 285
**15**	Kaempferol 3-*O*-rutinoside	5.15	264, 347	593	285
**16**	Quercetin 3-*O*-glucuronide-7-*O*-rhamnoside	5.24	253, 353	623	477, 301
**17**	Isorhamnetin 3-*O*-rutinoside	5.33	255, 354	623	315
**18**	Quercetin 3-*O*-glucuronide	5.53	255, 355	477	301

**Table 2 molecules-26-00997-t002:** Individual phenolic compounds (mg/L) identified by ultra-performance reverse-phase liquid chromatography (UPLC−PDA−MS/MS) in *Primula veris* L. flower extracts.

Compound	Water 100 °C				Ethanol 40% (*v*/*v*)				Ethanol 70% (*v*/*v*)				Ethanol 96% (*v*/*v*)			
	Fresh	Dried1	Dried2	Dried3	Fresh	Dried1	Dried2	Dried3	Fresh	Dried1	Dried2	Dried3	Fresh	Dried1	Dried2	Dried3
**1**	2.77 ± 0.06	14.49 ± 1.31	18.09 ± 0.04	25.76 ± 0.46	tr	12.10 ± 0.23	tr	22.72 ± 0.92	1.55 ± 0.06	7.05 ± 0.30	25.73 ± 0.71	tr	1.74 ± 0.02	tr	10.98 ± 0.13	21.72 ± 2.24
**2**	2.39 ± 0.03	22.49 ± 0.05	28.85 ± 1.06	52.30 ± 0.99	tr	25.09 ± 0.96	tr	35.83 ± 1.89	2.71 ± 0.14	13.52 ± 0.83	64.96 ± 10.20	tr	2.97 ± 0.11	tr	29.70 ± 0.13	43.02 ± 1.23
**3**	tr	349.92 ± 12.90	tr	tr	tr	321.30 ± 15.66	tr	tr	tr	369.00 ± 10.25	tr	tr	tr	115.40 ± 6.42	tr	tr
**4**	46.63 ± 4.22	259.48 ± 0.30	152.46 ± 15.73	169.68 ± 8.27	26.07 ± 2.18	270.99 ± 3.23	144.55 ± 19.44	213.83 ± 24.62	30.21 ± 3.48	355.33 ± 55.81	271.46 ± 21.58	195.56 ± 10.88	27.16 ± 0.51	223.12 ± 9.38	107.73 ± 4.10	164.84 ± 27.78
**5**	tr	86.38 ± 8.91	tr	tr	tr	87.71 ± 0.40	tr	tr	tr	109.03 ± 1.39	tr	tr	tr	44.18 ± 3.35	tr	tr
**6**	tr	208.79 ± 5.98	tr	tr	tr	216.97 ± 9.32	355.19 ± 14.92	tr	tr	334.55 ± 26.60	tr	576.34 ± 43.77	tr	178.22 ± 1.34	tr	tr
**7**	tr	338.78 ± 7.24	tr	tr	tr	329.02 ± 12.52	tr	tr	tr	381.15 ± 34.71	tr	35.72 ± 0.23	tr	125.74 ± 2.34	tr	tr
**8**	11.91 ± 1.23	76.18 ± 12.84	34.80 ± 3.67	35.10 ± 1.34	7.00 ± 0.29	83.92 ± 3.27	24.25 ± 0.18	29.84 ± 4.69	6.25 ± 0.98	120.21 ± 16.56	47.37 ± 1.38	tr	5.35 ± 0.02	65.38 ± 0.65	20.41 ± 0.32	43.04 ± 1.85
**9**	27.34 ± 0.78	tr	56.66 ± 5.48	114.74 ± 4.48	23.21 ± 1.76	Tr	58.65 ± 1.09	128.33 ± 1.64	19.13 ± 0.24	tr	127.84 ± 21.13	99.00 ± 0.98	17.11 ± 0.74	tr	42.54 ± 2.81	114.00 ± 2.17
**10**	205.96 ± 4.40	239.80 ± 23.20	508.64 ± 11.79	964.03 ± 7.96	152.98 ± 1.15	264.04 ± 4.78	493.66 ± 4.88	1000.54 ± 79.55	158.28 ± 12.59	308.79 ± 8.98	804.25 ± 29.50	907.07 ± 49.49	133.27 ± 5.07	242.90 ± 2.74	347.93 ± 8.09	879.66 ± 30.79
**11**	3.58 ± 0.60	58.02 ± 1.34	10.14 ± 0.44	26.84 ± 0.49	2.38 ± 0.04	51.75 ± 0.81	tr	12.70 ± 1.16	2.24 ± 0.20	74.54 ± 12.32	14.75 ± 1.13	6.24 ± 0.07	1.85 ± 0.07	42.06 ± 2.75	11.09 ± 1.12	16.99 ± 1.07
**12**	3.45 ± 0.36	40.53 ± 1.74	65.87 ± 1.25	195.93 ± 3.06	1.90 ± 0.02	71.02 ± 4.70	167.78 ± 1.89	199.82 ± 27.53	5.06 ± 0.70	93.91 ± 3.44	180.87 ± 9.32	257.70 ± 16.86	3.70 ± 0.03	103.66 ± 2.35	75.42 ± 6.22	78.88 ± 8.36
**13**	0.96 ± 0.09	8.47 ± 0.16	11.60 ± 0.41	16.70 ± 1.10	0.61 ± 0.03	8.43 ± 0.20	10.89 ± 0.71	9.20 ± 0.83	0.64 ± 0.06	8.84 ± 0.68	17.71 ± 0.19	10.15 ± 0.23	0.31 ± 0.01	7.29 ± 0.32	9.97 ± 0.81	8.58 ± 0.41
**14**	11.19 ± 0.26	8.47 ± 0.30	22.45 ± 1.41	36.04 ± 0.84	10.51 ± 0.12	4.26 ± 0.43	14.22 ± 0.32	28.36 ± 0.83	8.66 ± 0.25	4.47 ± 0.23	38.53 ± 5.62	23.02 ± 1.02	7.17 ± 0.11	3.30 ± 0.26	13.75 ± 0.24	37.78 ± 1.14
**15**	27.59 ± 1.18	21.87 ± 1.38	72.75 ± 7.71	141.81 ± 14.29	23.60 ± 1.54	26.75 ± 2.21	83.14 ± 3.69	174.60 ± 28.87	24.55 ± 4.06	32.11 ± 0.35	139.00 ± 20.74	154.05 ± 12.28	18.52 ± 1.22	29.66 ± 1.04	50.02 ± 1.66	134.63 ± 2.43
**16**	47.44 ± 0.90	35.06 ± 3.71	95.03 ± 4.58	167.21 ± 13.79	30.65 ± 0.70	36.07 ± 2.94	51.62 ± 4.11	178.12 ± 6.53	29.22 ± 1.07	49.77 ± 7.25	158.03 ± 9.76	112.98 ± 3.95	25.03 ± 0.58	36.11 ± 2.54	64.12 ± 0.48	171.64 ± 0.69
**17**	230.03 ± 8.05	88.20 ± 4.25	521.76 ± 15.79	1029.3 ± 83.95	164.26 ± 7.28	111.41 ± 1.92	481.51 ± 16.85	1127.71 ± 86.66	171.35 ± 13.17	143.41 ± 21.40	883.34 ± 77.32	1001.11 ± 70.36	131.49 ± 13.25	119.59 ± 11.51	342.23 ± 14.06	1035.57 ± 25.44
**18**	7.01 ± 0.44	13.52 ± 0.41	88.37 ± 1.59	161.07 ± 2.77	7.12± 0.57	33.07 ± 1.09	169.44 ± 11.91	161.62 ± 8.33	10.04 ± 0.52	44.79 ± 2.77	224.01 ± 27.23	212.87 ± 20.49	7.15 ± 0.59	51.87 ± 2.97	73.09 ± 3.74	93.36 ± 0.71
**Total**	628.24 ± 19.24	1870.46 ± 47.02	1687.45 ± 24.49	3136.59 ± 121.53	450.28 ± 1.50	1953.90 ± 20.13	2054.89 ± 18.59	3323.22 ± 250.10	469.90 ± 35.57	2450.46 ± 183.20	2997.85 ± 214.44	3591.82 ± 94.74	382.84 ± 8.48	1388.47 ± 40.71	1198.98 ± 19.53	2843.71 ± 83.12

Fresh: fresh flowers; Dried1: commercial sample; Dried2: dried flowers collected in 2018; Dried3: dried flowers collected in 2019; Mean values ± SD (*n* = 3); tr: traces under LOD (limit of detection).

**Table 3 molecules-26-00997-t003:** Individual phenolic compounds (mg/L) identified by ultra-performance reverse-phase liquid chromatography (UPLC−PDA−MS/MS) in *Primula veris* L. flowers with stalks extracts.

Compound	Water 100 °C		Ethanol 40% (*v*/*v*)		Ethanol 70% (*v*/*v*)		Ethanol 96% (*v*/*v*)	
	Fresh	Dried	Fresh	Dried	Fresh	Dried	Fresh	Dried
**1**	1.15 ± 0.01	17.27 ± 0.17	tr	tr	1.08 ± 0.04	17.23 ± 0.55	0.76 ± 0.04	16.51 ± 0.29
**2**	0.95 ± 0.01	37.92 ± 3.43	tr	tr	1.15 ± 0.02	25.45 ± 2.93	0.45 ± 0.01	28.24 ± 0.53
**3**	tr	tr	tr	tr	tr	tr	tr	tr
**4**	17.65 ± 0.04	168.94 ± 6.23	16.56 ± 2.23	95.21 ± 5.30	14.94 ± 0.57	122.79 ± 7.52	14.38 ± 0.62	101.37 ± 4.94
**5**	tr	tr	tr	tr	tr	tr	tr	tr
**6**	tr	tr	tr	204.67 ± 15.54	tr	tr	tr	tr
**7**	tr	tr	tr	tr	tr	tr	tr	tr
**8**	5.74 ± 0.16	43.13 ± 0.92	tr	17.00 ± 0.32	3.24 ± 0.14	34.06 ± 2.71	4.24 ± 0.05	27.11 ± 1.03
**9**	13.42 ± 0.29	75.20 ± 12.68	25.86 ± 0.48	43.47 ± 0.43	12.67 ± 0.48	79.39 ± 7.23	16.94 ± 1.35	54.83 ± 2.14
**10**	73.83 ± 12.44	519.73 ± 54.75	40.64 ± 0.40	385.78 ± 21.05	71.91 ± 2.80	540.63 ± 74.48	45.89 ± 4.18	389.76 ± 3.22
**11**	1.34 ± 0.14	28.24 ± 2.73	0.52 ± 0.03	tr	1.08 ± 0.01	9.52 ± 0.85	0.84 ± 0.12	17.15 ± 0.31
**12**	0.82 ± 0.08	47.90 ± 1.11	1.23 ± 0.01	113.22 ± 7.41	1.98 ± 0.04	86.42 ± 2.51	1.12 ± 0.10	69.12 ± 1.08
**13**	0.16 ± 0.00	18.00 ± 0.77	0.28 ± 0.02	4.72 ± 0.11	0.13 ± 0.00	4.06 ± 0.67	0.01 ± 0.00	10.38 ± 0.69
**14**	2.74 ± 0.12	38.73 ± 0.74	9.53 ± 0.22	11.88 ± 0.53	4.35 ± 0.29	29.37 ± 1.08	4.20 ± 0.69	19.85 ± 0.46
**15**	12.11 ± 0.23	75.35 ± 2.64	2.60 ± 0.12	57.13 ± 4.55	10.60 ± 0.25	82.62 ± 6.35	6.27 ± 0.23	55.97 ± 5.64
**16**	21.02 ± 0.74	167.94 ± 10.57	23.50 ± 1.87	61.04 ± 2.14	16.72 ± 1.68	130.51 ± 6.72	22.52 ± 1.73	100.76 ± 8.31
**17**	105.56 ± 6.64	730.95 ± 77.44	53.96 ± 1.89	450.43 ± 31.66	84.56 ± 6.97	592.25 ± 6.45	57.08 ± 2.94	431.77 ± 35.21
**18**	3.27 ± 0.35	29.96 ± 1.44	5.16 ± 0.36	109.06 ± 10.50	3.96 ± 0.32	103.08 ± 15.02	3.91 ± 0.04	59.57 ± 1.03
**Total**	259.77 ± 6.76	1999.26 ± 16.86	179.85 ± 5.88	1553.61 ± 41.91	228.37 ± 11.86	1857.36 ± 100.46	178.61 ± 4.88	1382.41 ± 54.39

Mean values ± SD (*n* = 3); tr: traces under LOD (limit of detection).

**Table 4 molecules-26-00997-t004:** Individual phenolic compounds (mg/L) identified by ultra-performance reverse-phase liquid chromatography (UPLC−PDA−MS/MS) in *Primula veris* L. stalk extracts.

Compound	Water 100 °C		Ethanol 40% (*v*/*v*)		Ethanol 70% (*v*/*v*)		Ethanol 96% (*v*/*v*)	
	Fresh	Dried	Fresh	Dried	Fresh	Dried	Fresh	Dried
**1**	tr	tr	tr	tr	tr	tr	tr	tr
**2**	1.23 ± 0.02	tr	tr	tr	tr	tr	tr	tr
**3**	tr	tr	tr	tr	tr	tr	tr	tr
**4**	6.66 ± 0.60	17.84 ± 0.99	5.55 ± 0.21	35.00 ± 0.66	5.89 ± 0.47	50.37 ± 7.91	4.03 ± 0.31	34.38 ± 0.08
**5**	tr	tr	tr	tr	tr	tr	tr	tr
**6**	tr	tr	tr	tr	tr	tr	tr	tr
**7**	tr	tr	tr	tr	tr	tr	tr	tr
**8**	2.92 ± 0.30	7.17 ± 0.13	1.82 ± 0.03	11.16 ± 0.05	2.20 ± 0.06	15.95 ± 2.20	tr	13.37 ± 0.38
**9**	6.04 ± 0.17	23.32 ± 0.23	7.04 ± 0.47	41.89 ± 1.80	8.11 ± 1.34	56.22 ± 5.05	8.39 ± 0.09	22.36 ± 0.48
**10**	11.56 ± 0.25	81.25 ± 4.43	21.09 ± 0.49	142.24 ± 5.41	18.56 ± 0.68	166.90 ± 4.85	5.79 ± 0.38	80.38 ± 13.55
**11**	tr	2.16 ± 0.02	0.20 ± 0.02	4.02 ± 0.16	0.16 ± 0.01	5.90 ± 0.98	tr	1.71 ± 0.18
**12**	tr	0.23 ± 0.01	tr	tr	tr	0.51 ± 0.02	tr	tr
**13**	tr	1.39 ± 0.03	tr	1.73 ± 0.03	tr	1.79 ± 0.14	tr	1.26 ± 0.03
**14**	3.05 ± 0.07	9.57 ± 0.42	1.46 ± 0.03	14.96 ± 0.23	1.23 ± 0.19	20.01 ± 1.03	4.59 ± 0.16	10.92 ± 0.47
**15**	2.52 ± 0.11	14.09 ± 1.12	1.90 ± 0.06	19.14 ± 1.27	1.50 ± 0.22	24.49 ± 0.27	16.43 ± 1.15	9.66 ± 0.18
**16**	17.10 ± 0.33	76.41 ± 2.67	17.41 ± 0.13	102.55 ± 2.38	19.39 ± 1.20	153.55 ± 22.38	tr	68.09 ± 2.38
**17**	20.55 ± 0.72	213.94 ± 15.03	36.30 ± 1.49	239.62 ± 24.14	39.94 ± 3.50	390.62 ± 58.30	tr	180.19 ± 11.34
**18**	tr	tr	tr	tr	tr	tr	tr	tr
**Total**	71.46 ± 1.74	447.36 ± 7.28	92.78 ± 1.52	612.29 ± 15.62	97.07 ± 7.55	886.31 ± 91.34	39.23 ± 1.77	422.31 ± 5.97

Mean values ± SD (*n* = 3); tr: traces under LOD (limit of detection).

**Table 5 molecules-26-00997-t005:** Individual phenolic compounds (mg/L) identified by ultra-performance reverse-phase liquid chromatography (UPLC−PDA−MS/MS) in *Primula veris* L. leaf extracts.

Compound	Water 100 °C		Ethanol 40% (*v*/*v*)		Ethanol 70% (*v*/*v*)		Ethanol 96% (*v*/*v*)	
	Fresh	Dried	Fresh	Dried	Fresh	Dried	Fresh	Dried
**1**	2.98 ± 0.27	tr	tr	tr	tr	tr	2.43 ± 0.13	tr
**2**	tr	tr	tr	tr	tr	tr	tr	tr
**3**	tr	tr	tr	tr	tr	tr	tr	tr
**4**	57.29 ± 0.07	268.41 ± 7.69	43.91 ± 5.90	252.69 ± 1.14	53.52 ± 0.24	275.79 ± 7.66	42.04 ± 1.81	98.65 ± 2.80
**5**	tr	tr	tr	tr	tr	tr	tr	tr
**6**	6.49 ± 0.19	tr	tr	tr	tr	tr	tr	tr
**7**	tr	26.62 ± 2.80	6.32 ± 0.48	19.56 ± 0.76	tr	23.07 ± 1.83	6.80 ± 1.07	10.14 ± 0.04
**8**	89.98 ± 15.17	364.51 ± 35.26	92.07 ± 0.69	377.42 ± 3.12	94.31 ± 0.78	425.56 ± 38.75	80.64 ± 1.03	184.11 ± 1.72
**9**	tr	tr	tr	tr	tr	tr	tr	tr
**10**	56.46 ± 5.46	255.98 ± 10.98	62.03 ± 0.61	277.86 ± 4.35	64.74 ± 1.01	319.64 ± 28.70	57.86 ± 5.27	125.20 ± 3.35
**11**	1.99 ± 0.05	7.92 ± 0.15	2.55 ± 0.14	8.15 ± 0.54	2.14 ± 0.14	8.78 ± 0.26	2.35 ± 0.32	5.05 ± 0.03
**12**	tr	tr	tr	tr	tr	tr	tr	tr
**13**	tr	tr	tr	tr	tr	0.55 ± 0.02	tr	tr
**14**	12.30 ± 0.43	43.82 ± 4.64	14.36 ± 0.33	43.27 ± 3.57	12.10 ± 1.00	50.91 ± 3.91	14.41 ± 2.38	29.89 ± 0.65
**15**	19.78 ± 1.24	68.78 ± 3.31	26.02 ± 1.15	80.23 ± 6.54	29.18 ± 2.38	90.95 ± 4.69	23.75 ± 0.87	49.42 ± 1.92
**16**	16.39 ± 1.74	65.21 ± 1.97	16.87 ± 1.34	65.92 ± 1.14	16.26 ± 0.28	77.47 ± 0.84	15.68 ± 1.20	34.09 ± 0.59
**17**	38.57 ± 1.86	210.02 ± 3.79	51.42 ± 1.80	206.41 ± 6.83	46.87 ± 1.55	283.48 ± 41.31	40.40 ± 2.08	102.90 ± 3.71
**18**	tr	tr	tr	tr	tr	tr	tr	tr
**Total**	302.22 ± 20.41	1311.28 ± 61.01	315.55 ± 10.11	1331.51 ± 27.99	319.14 ± 5.83	1556.21 ± 118.09	286.36 ± 8.13	639.45 ± 1.79

Mean values ± SD (*n* = 3); tr: traces under LOD (limit of detection).

**Table 6 molecules-26-00997-t006:** Individual phenolic compounds (mg/L) identified by ultra-performance reverse-phase liquid chromatography (UPLC−PDA−MS/MS) in *Primula veris* L. root extracts.

Compound	Water 100 °C		Ethanol 40% (*v*/*v*)		Ethanol 70% (*v*/*v*)		Ethanol 96% (*v*/*v*)	
	Fresh	Dried	Fresh	Dried	Fresh	Dried	Fresh	Dried
**1**	tr	0.36 ± 0.34	tr	0.59 ± 0.58	tr	tr	tr	0.68 ± 0.04
**2**	tr	tr	1.09 ± 0.06	tr	6.35 ± 1.00	25.72 ± 0.70	tr	tr
**3**	tr	tr	tr	tr	0.22 ± 0.04	0.96 ± 0.05	tr	tr
**4**	1.19 ± 0.06	3.34 ± 0.30	2.51 ± 0.07	4.91 ± 0.30	tr	tr	0.97 ± 0.10	4.06 ± 0.30
**5**	tr	tr	tr	tr	tr	tr	tr	tr
**6**	tr	0.30 ± 0.28	tr	0.43 ± 0.40	tr	tr	tr	0.58 ± 0.02
**7**	tr	tr	tr	tr	10.12 ± 1.59	41.93 ± 4.32	tr	tr
**8**	1.60 ± 0.16	19.80 ± 2.55	1.80 ± 0.12	29.16 ± 4.59	tr	tr	0.52 ± 0.04	20.30 ± 1.83
**9**	0.59 ± 0.05	3.42 ± 0.10	0.87 ± 0.05	5.02 ± 0.00	0.34 ± 0.05	1.96 ± 0.10	1.11 ± 0.01	2.18 ± 0.01
**10**	5.20 ± 0.11	11.78 ± 0.68	4.72 ± 0.01	17.32 ± 1.50	1.38 ± 0.22	6.21 ± 0.09	1.68 ± 0.04	13.36 ± 0.27
**11**	tr	0.24 ± 0.23	tr	0.35 ± 0.32	2.23 ± 0.35	9.24 ± 0.07	tr	1.66 ± 0.26
**12**	tr	tr	tr	tr	1.84 ± 0.29	17.45 ± 0.81	tr	tr
**13**	tr	tr	tr	tr	3.44 ± 0.54	15.28 ± 0.00	tr	tr
**14**	1.73 ± 0.04	3.18 ± 0.45	1.62 ± 0.03	4.68 ± 0.79	tr	tr	0.58 ± 0.00	3.20 ± 0.33
**15**	0.58 ± 0.03	1.87 ± 0.12	0.69 ± 0.01	2.75 ± 0.09	0.64 ± 0.10	2.39 ± 0.07	0.17 ± 0.01	2.72 ± 0.53
**16**	2.47 ± 0.27	9.80 ± 0.19	1.05 ± 0.03	14.41 ± 0.69	tr	tr	3.31 ± 0.00	11.98 ± 0.54
**17**	4.64 ± 0.16	7.17 ± 0.08	6.17 ± 0.02	10.54 ± 0.42	tr	tr	1.51 ± 0.00	17.85 ± 1.86
**18**	tr	0.13 ± 0.01	tr	0.20 ± 0.03	0.28 ± 0.04	1.30 ± 0.03	tr	0.16 ± 0.01
**Total**	30.32 ± 0.64	61.38 ± 2.60	20.50 ± 0.12	90.29 ± 6.42	26.85 ± 4.22	122.46 ± 4.23	9.86 ± 0.13	73.60 ± 3.03

Mean values ± SD (*n* = 3); tr: traces under LOD (limit of detection).

**Table 7 molecules-26-00997-t007:** Influence of plant parts of fresh *Primula veris* L. and extraction solution on the total polyphenolic compounds and antioxidant properties in the extracts.

		Plant Part					Extraction Solution			
							Water 100 °C	Ethanol (%) (*v*/*v*)		
		Flowers	Flowers and Stalks	Stalks	Leaves	Roots		40	70	96
**mg/L**	**Total polyphenols**	484.42 ^A^	211.07 ^C^	75.19 ^D^	306.83 ^B^	22.05 ^E^	259.55 ^A^	211.95 ^C^	229.23 ^B^	178.92 ^D^
**mg GAE/L**	**TPC**	492.05 ^A^	306.23 ^B^	145.56 ^C^	284.38 ^B^	23.41 ^D^	271.69 ^Ba^	247.71 ^Bb^	312.26 ^A^	169.63 ^C^
**mmol TE/L**	**DPPH**	2.31 ^B^	1.59 ^C^	0.66 ^D^	2.44 ^A^	0.21 ^E^	1.14 ^C^	2.00 ^A^	1.59 ^B^	1.04 ^D^
	**FRAP**	2.70 ^A^	1.57 ^C^	0.82 ^D^	2.23 ^B^	0.16 ^E^	1.45 ^C^	1.78 ^A^	1.65 ^B^	1.10 ^D^
	**ABTS**	4.53 ^A^	2.70 ^C^	1.15 ^D^	4.41 ^B^	0.41 ^E^	2.08 ^C^	3.58 ^A^	3.08 ^B^	1.82 ^D^

Statistically significant differences between means (^A–E^ for *p* ≤ 0.01; ^a,b^ for *p* ≤ 0.05), marked by a different letter in the rows.

**Table 8 molecules-26-00997-t008:** Influence of plant parts of dried *Primula veris* L. and extraction solution on the total polyphenolic compounds and antioxidant properties in extracts.

		**Plant Part**					**Extraction Solution**			
							**Water 100 °C**	**Ethanol (%) (*v*/*v*)**		
		**Flowers**	**Flowers and Stalks**	**Stalks**	**Leaves**	**Roots**		**40**	**70**	**96**
**mg/L**	**Total** **polyphenols**	2609.40 ^A^	1695.80 ^B^	595.41 ^D^	1132.30 ^C^	87.49 ^E^	1244.70 ^C^	1259.01 ^B^	1549.10 ^A^	843.68 ^D^
**mg GAE/L**	**TPC**	2190.10 ^A^	1432.40 ^B^	737.24 ^C^	1408.30 ^B^	93.81 ^D^	1156.70 ^C^	1309.30 ^B^	1512.40 ^A^	711.02 ^D^
**mmol TE/L**	**DPPH**	8.82 ^A^	5.67 ^B^	3.22 ^D^	5.49 ^C^	0.59 ^E^	4.12 ^C^	4.91 ^B^	6.01 ^A^	3.41 ^D^
	**FRAP**	12.49	7.11	4.17	7.38	9.16	5.63	6.97	8.22	11.42
	**ABTS**	15.37 ^A^	10.87 ^Ba^	6.02 ^C^	9.09 ^Bb^	1.12 ^D^	7.57 ^C^	9.15 ^B^	10.73 ^A^	6.52 ^C^

Statistically significant differences between means (^A–E^ for *p* ≤ 0.01; ^a,b^ for *p* ≤ 0.05), marked by a different letter in the rows.

## Data Availability

The data presented in this study are available on request from the corresponding author.

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
