# Peer review of "Bioactive Phenolic Compounds from Primula veris L.: Influence of the Extraction Conditions and Purification"

_molecules, 2021, doi:10.3390/molecules26040997_

Round 1
Reviewer 1 Report
Minor remarks
Use italic letters for Greek alphabet (e.g., alpha, beta, gamma, Latin names of plants).
Blank should be provided between quantity and unit (for instance, Celsius degree).
Provide information for ethanol concentrations, the volumetric or weight percentages.
The line space is not the same through the whole manuscript.
Citing the references is not allowed in the Conclusion.
Pages 4, 6, 8, 9, 10, 12, 13, 14, 15, 16…These pages have a logo of the journal. I think that it occurred due to inserted section brake. Also, the order of the pages is not adequate.
All other minor comments are given in the document.
Major remarks
Also, the part which refers to the application of advanced extraction techniques should be improved and better described. The following references are desirable to include in the manuscript: DOI: 10.3390/antiox8080248, DOI: 10.1007/s13197-020-04312-w.
It is desirable to depict the chromatogram of the extract that has the highest content of polyphenols and mark the peaks of identified compounds.

Author Response
Dear Reviewer,
First of all, we would like to express our sincere gratitude for the revision of the manuscript. We greatly appreciate your work. We considered and accepted all suggestions and comments.
Point 1: All necessary corrections have been added to the text and marked in yellow.
Response 1: We made changes and corrections in the text according to Your recommendations.
Point 2: Use italic letters for Greek alphabet (e.g., alpha, beta, gamma, Latin names of plants).
Response 2: It has been corrected.
Point 3: Blank should be provided between quantity and unit (for instance, Celsius degree).
Response 3: It has been corrected.
Point 4: Provide information for ethanol concentrations, the volumetric or weight percentages.
Response 4: It has been completed.
Point 5: The line space is not the same through the whole manuscript.
Response 5: It has been improved.
Point 6: Citing the references is not allowed in the Conclusion.
Response 6: It has been removed.
Point 7: Pages 4, 6, 8, 9, 10, 12, 13, 14, 15, 16…These pages have a logo of the journal. I think that it occurred due to inserted section brake. Also, the order of the pages is not adequate.
Response 7: It is because now Molecules has the template file. We had big tables and it was very difficult to fit them into the template. It will be improved in the final editorial version.
Point 8: Also, the part which refers to the application of advanced extraction techniques should be improved and better described. The following references are desirable to include in the manuscript: DOI: 10.3390/antiox8080248, DOI: 10.1007/s13197-020-04312-w.
Response 8: It has been modified in the paragraph of discussion in the lines 476-484.
Point 9: It is desirable to depict the chromatogram of the extract that has the highest content of polyphenols and mark the peaks of identified compounds.
Response 9: It has been added in the lines 143-144.
Reviewer 2 Report
L. stands for Line
Main comments in this report are about the experimental design and the statistical analysis:
a) In this work several origin of samples are indicated that I will refer as follows
a1: two different crops: 2018 and 2019 (L. 573)
a2: Commercial dried herbs (n=3) (L.587)
a3: in L. 575 test material (maybe both crops in a1 or only the sooner crop) seems to be the basis for the experimental research:
b) In Table 2 three different dried samples are shown (for flowers) whereas in tables 3 to 7 only one dried sample is studied. Please, clarify why?.
c) In L.149 to L.151 authors say that cowslip collected in 2019 provides the most valuable extracts. But, in L.573 only two crops are indicated, why 3 columns?
d) Observe that not statistical analysis has been made to study differences on time.
e) Commercial dried herbs (a2; L.587): where do they appear in this study maybe in Table 2?, if so, where do the two different crops (a1, L.573) appear?
f) Anova two way (L670) is used for the statistical analysis [one for fresh , Table 7; another one for dry; Table 8]. So that, two factors are considered: F1:morphological part of the plant and F2:solvent. Authors do not indicate what happens with the interaction.
g) Table 8: FRAP method does not entail differences among treatments? Please, explain this.
h) Authors say that Pearson correlation test is used in the analysis (L.674 to 676); where? Although numerical results can be disregarded, conclusions about this analysis should be written.
i) Numerical basis for the heatmap is not indicated. Observe that heatmap can put colours based on internal references (darkest red color can correspond with the highest value obtained in the sample, but may not be a large value in the population), this is why the reader needs a short explanation of the numbers behind the colors
j)In L.666 authors say that analyses were made with 3 independent replications for each sample. A sample is a plant harvested in 2019? Please, clarify which are the inputs in the experiment taking observation a) above into consideration.
k) If the same plant is considered to measure different morphological parts, dependent observation appear in the study, would this change your statistical technique for the analysis of observations?
Minor comments:
L46 a reference is needed to justify the use of saponins in cancer research, is a topic of general interest that deserves scientific support.
L.671: p<0.01 and p<0.05
small things: : question tag L53,L73 should be rephrased, articles in L.74, 75, and in other sentences should be revised.
Author Response
Dear Reviewer,
First of all, we would like to express our sincere gratitude for the revision of the manuscript. We greatly appreciate your work. We considered and accepted all suggestions and comments, and also made changes and corrections in the text according to Your recommendations.
Point 1: b) In Table 2 three different dried samples are shown (for flowers) whereas in tables 3 to 7 only one dried sample is studied. Please, clarify why?
Response 1: In the presented research, we wanted to compare the extraction efficiency depending on the solvent used as well as the morphological part of the plant. For comparison, we wanted to use plant material not only collected by us from natural habitat but also the commercial samples available for anybody by all the year. Additionally, according to our knowledge from our preliminary study, the richest source of phenolic compounds are flowers of Primula veris L., that’s why we chose flowers for the comparison in three options as dried and one fresh. The commercial samples of cowslip containing a different morphological part are not available apart from the flowers and roots. In tables 3 to 7 the mean results for three independent extractions for dried and fresh morphological part of the plant, collected in the year 2019 were presented.
Point 2: c) In L.149 to L.151 authors say that cowslip collected in 2019 provides the most valuable extracts. But, in L.573 only two crops are indicated, why 3 columns?
Response 2: In the first column the results obtained for the commercial sample were presented (Dried 1), in the second and the third: the results for dried flowers obtained from natural habitat (Dried 2- dried flowers collected in 2018, Dried3- dried flowers collected in 2019)- the explanations of abbreviations is presented below the table 2. In lines, L.149 to L.151, the comparison of the extraction efficiency dependent on the harvest year (between sample Dried2 and Dried3) was presented to show that how changes in the content of polyphenolic compounds occurred after year they storage.
Point 3: d) Observe that not statistical analysis has been made to study differences on time.
Response 3: Changes over time were not statistically analyzed because, firstly, it was not important for us at the extraction optimization stage, and secondly, the morphological parts of the plant were collected and examined only in the year 2019. The analysis of the polyphenolic activity of cowslip extracts over time will be much more valuable for our research project depending on their storage conditions, to obtain data on their stability due to the fact of potential use as a food additive
Point 4: e) Commercial dried herbs (a2; L.587): where do they appear in this study maybe in Table 2?, if so, where do the two different crops (a1, L.573) appear?
Response 4: This was explained in point a). and also the paragraph mentioned in the question was re-edited and the literature item referred to in the context of the sentence was completed. In the current version it reads as follows:
Bearing in our previous study including the extraction from dried tea [43] and the high heterogeneity of the tested dried plant material, a range of 6 to 8% of the water content was assumed to be sufficient to stabilize the test material and complete the drying process.
Point 5: f) Anova two way (L670) is used for the statistical analysis [one for fresh , Table 7; another one for dry; Table 8]. So that, two factors are considered: F1:morphological part of the plant and F2:solvent. Authors do not indicate what happens with the interaction.
Response 5: We did not analyze the interaction due to the volume of our manuscript. We focused on these two factors. The statistical tables have been separated for fresh material, characterized by significantly lower concentrations and values of the tested parameters, and for dried material, to facilitate interpretation and maintain clarity of study.
Point 6: g) Table 8: FRAP method does not entail differences among treatments? Please, explain this.
Response 6: The antioxidant activity measured by FRAP method is based on the reduction of Fe3+ to Fe2+ (for instance thanks to that antioxidants can behave as prooxidants- antioxidants might make these reduced forms available to reduce hydrogen peroxide or organic hydroperoxides in a Fenton-type reaction, thereby increasing the rate of initiation). It was primarily used to determine the reducing power of serum or plasma, but recently, it has also been adapted to the research of plant-derived antioxidants. This method has many advantages (such as quick and simple to perform and can be easily automated, highly reproducible over a wide concentration range) but it is a spectrophotometric method and interference may occur, what can cause overstated results. We checked the differences between extracts obtained by a different type of extraction using ANOVA followed by Duncan’s multiple test. Before applying the test, we checked whether we could apply this test for this data (we checked if the distribution is normal and what is the distribution of variance). The results showed there are no significant differences among treatments.
Point 7: h) Authors say that Pearson correlation test is used in the analysis (L.674 to 676); where? Although numerical results can be disregarded, conclusions about this analysis should be written.
Response 7: Thank you for the correction, we forgot to describe the Pearson correlation between tested parameters. The information was added in lines 335-343 and also I have presented this paragraph below:
Antioxidant capacity of the tested extracts was highly correlated with the content of the phenolic content obtained both by TPC and HPLC regardless of the extraction method used: TPC vs DPPH r= 0.961; TPC vs FRAP 0.974; TPC vs ABTS= 0.937; HPLC analysis vs DPPH r= 0.929, vs FRAP= 0.955; vs ABTS= 0.941. Which confirms previous observations that phenolic compounds are largely responsible for the antioxidant activity. High correlation between TPC and total phenolic content obtained by HPLC method was observed (r= 0.94), as well as between antioxidant activity measurement by different test (DPPH vs FRAP r= 0.961; DPH vs ABTS r= 0.977; ABTS vs FRAP r= 0.947). Which confirms the correctness of the methods performed.
Point 8: i) Numerical basis for the heatmap is not indicated. Observe that heatmap can put colours based on internal references (darkest red color can correspond with the highest value obtained in the sample, but may not be a large value in the population), this is why the reader needs a short explanation of the numbers behind the colors
Response 8: In the cluster analysis, when we have data with the different units we need to standardized (or normalized) the data to get proper results of the analysis. To standardize variables the program (Statistica) calculate the mean and standard deviation for a variable. By transforming each result obtained in the measurement by applying the standardization formula, we obtain a normalized measure, where the expected value (mean) is 0 and the variance is 1. Thanks to this, we can determine how far a given result (x) is from the mean value in the language statistical. Z = 1 means that the given result is higher than the mean by 1 standard deviation. Z = -0.5 means that the given result is lower than the mean by 0.5 standard deviation. This is a mathematical calculation. In this analysis, we are not focusing on the exact value of each data. The reason this importance is particularly high in cluster analysis is that groups are defined based on the distance between points in mathematical space. Heatmap is only a graphical visualization. We hope that adding information that the data has been standardized in the text and under the figure will be sufficient.
The missing information and some correction were added under figure 1: The red colour on the heat map means the highest content of a particular phenol compound or the highest antioxidant activity, whereas the dark green means the low value of these parameters. Cluster analysis is performed using standardized data. The correction was also made in the text (lines 353-357 and lines 382-385). The analyzed variables had a different unit, so the standardization of values was made. Finally, a color scheme (heatmap) is applied for the visualization and the data matrix is displayed. Based on the color scale in the heatmap, the values of the individual parameters can be compared (the darkest read means the highest value of the given compound content or antioxidant activity, whereas the darkest green means the lowest value).
Point 9: j) In L.666 authors say that analyses were made with 3 independent replications for each sample. A sample is a plant harvested in 2019? Please, clarify which are the inputs in the experiment taking observation a) above into consideration.
Response 9: In the year 2018, based on the conducted experiments with fresh flowers of Primula veris L., a very high antioxidant potential was demonstrated in this plant. In the next harvest season (2019), other morphological parts of this plant were also analyzed to be sure that the other better sources of health-promoting substances not omitted. For this purpose, 3 independent samples were collected from the crop. Each sample contained at least 15 whole plants, which were cleaned and divided into morphological parts, and then each of them was divided into two parts as fresh material and material undergoing drying. Separate extraction and analyzes were performed for each of them, i.e. 3 independent extractions for each of the morphological parts in the dried and fresh version.
Bearing in mind that the commercially available plant material is a mixture of a given plant, not only within different crops but also varieties (Primula veris, Primula elatior), 3 independent extractions were done for each.
Point 10: k) If the same plant is considered to measure different morphological parts, dependent observation appear in the study, would this change your statistical technique for the analysis of observations?
Response 10:The morphological parts obtained from the plants were averaged over each of the 3 samples taken from the crop. The individual morphological parts within each plant were not analyzed. However, the differences between the morphological parts were statistically confirmed in Tables 7 and 8.
Point 11: L46 a reference is needed to justify the use of saponins in cancer research, is a topic of general interest that deserves scientific support.
Response 11: It has been completed in line 46.
Point 12: L.671: p<0.01 and p<0.05
Response 11: It has been completed in line 677.
Point 13: small things: : question tag L53,L73 should be rephrased, articles in L.74, 75, and in other sentences should be revised.
Response 12: The meaning of the suggestion was understood and the question mark was removed from line L53 as redundant in the context of the sentence.
The following lines, as suggested by the reviewer, were re-edited because the main goal was presented twice, while the first one was only one of the assumptions of the experiment.
Reviewer 3 Report
In the present study, the authors submitted a very interesting way to describe a different way of why is important to study and to know the phenolic compounds on different organs of a traditional medicinal plant as Primula veris. They propose simple methods and the use of GRAS solvent, focusing in maximise extraction efficiency and thus the concentration of polyphenolic compounds. However, there are some observations that need to attend.
Line 59: Please change “areas” for organs or plant parts. If they want to mention "areas" they have at least to refer as foliar area or root area.
Lines 179-182: How fast did the temperature low? How many degrees per second or minute in each case? Please be more explicit on this paragraph.
Lines 196-197: In the manuscript, there is no quantification of structural polysaccharides, so please mention the reference or the references which report this characteristic.
Lines 256-258: Please mention the references which report “It appears from scientific literature and pharmacopoeia papers that these underground parts of cowslip are used in pharmaceutical preparations with expectorant effect due to their high levels of saponins”.
Tables 7 and 8: Please correct mg/L: FRAP, DPPH, and ABTS their units are expressed as mmol of Trolox equivalents per 1 L, only is correct to TPC.
Lines 391-392: The authors mention “This activity is inversely proportional to the antioxidant properties that decrease in the presence of sugar substituents”, the literature that they refer do not say anything about this, please correct.
Author Response
Dear Reviewer,
first of all, we would like to express our sincere gratitude for the revision of the manuscript. We greatly appreciate your work. We considered and accepted all suggestions and comments, and also made changes and corrections in the text according to Your recommendations.
Point 1: Line 59: Please change “areas” for organs or plant parts. If they want to mention "areas" they have at least to refer as foliar area or root area.
Response 1: It has been changed for „plant parts”, in line 59.
Point 2: Lines 179-182: How fast did the temperature low? How many degrees per second or minute in each case? Please be more explicit on this paragraph.
Response 2: The final solution temperature before ultrasound-assisted extraction in the case of fresh material was 90-92 °C, while in the case of dried material the final temperature was 98 °C. The temperature drop during the entire extraction process was not monitored. Due to the homogeneity of the entire extraction process under the mildest possible temperature conditions, the temperature control of the samples during extraction was established at a safe level of 40 °C.
Point 3: Lines 196-197: In the manuscript, there is no quantification of structural polysaccharides, so please mention the reference or the references which report this characteristic.
Response 3: It has been completed in line 215.
Point 4: Lines 256-258: Please mention the references which report “It appears from scientific literature and pharmacopoeia papers that these underground parts of cowslip are used in pharmaceutical preparations with expectorant effect due to their high levels of saponins”.
Response 4: It has been completed in lines 270-271.
Point 5: Tables 7 and 8: Please correct mg/L: FRAP, DPPH, and ABTS their units are expressed as mmol of Trolox equivalents per 1 L, only is correct to TPC.
Response 5: It has been improved by adding the proper units in the table 7 and 8.
Point 6: Lines 391-392: The authors mention “This activity is inversely proportional to the antioxidant properties that decrease in the presence of sugar substituents”, the literature that they refer do not say anything about this, please correct.
Response 6: It has been changed for proper literature.
Round 2
Reviewer 2 Report
Authors have clearly explained most of the points indicated in the previous report. However, I would like to add four suggestions:
a) Following authors clarifications, for the ANOVA analysis only the test material from the 2019 harvest is used. If so, this would be helpful to add in section 4.7. Also, in 4.2 would be helpful to indicate that comercial and 2018 harvest material are only used for descriptive statistical comparisons for flowers in Table 2.
b) In a two way ANOVA interaction must be studied between both factors. If the interaction is significant, comparisons of the main factors do not make sense. If it is not significant, comparisons as those presented in Table 7 (or 8) are valid. Which is the case here?
c) After the correlation analysis added in the last version, it seems strange that if FRAP method is so correlated with ABTS or DPPH, there are no differences statistically significant in FRAP method. Did the researchers find a high variability in measurements with this method?
Author Response
Dear Reviewer,
We accept all suggestions and comments, and also made changes and corrections in the text according to Your recommendations. The responses to the suggestions are attached.
